# N-Doped Honeycomb-like Ag@N-Ti_3_C_2_T*_x_* Foam for Electromagnetic Interference Shielding

**DOI:** 10.3390/nano12172967

**Published:** 2022-08-27

**Authors:** Xiaohan Wang, Fan Zhang, Feiyue Hu, Yaya Li, Yongqiang Chen, Hailong Wang, Zhiyu Min, Rui Zhang

**Affiliations:** 1School of Material Science and Engineering, Zhengzhou University, Zhengzhou 450001, China; 2Henan Vocational College of Information and Statistics, Zhengzhou 450008, China; 3School of Material Science and Engineering, Luoyang Institute of Science and Technology, Luoyang 471026, China

**Keywords:** nitrogen doping, Ag@N-Ti_3_C_2_T*_x_* composites, honeycomb-like structure, electromagnetic interference, multi-reflections

## Abstract

To solve the pollution problem of electromagnetic waves, new electromagnetic shielding materials should meet the requirements of being lightweight with high electrical conductivity. In this work, the combination of silver (Ag) nanoparticles and nitrogen doping (N-doping) was expected to tune the electromagnetic and physical properties of Ti_3_C_2_T*_x_* MXene, and the Ag@N-Ti_3_C_2_T*_x_* composites were fabricated through the hydrothermal reactions. The nitrogen doped (N-doped) Ag@Ti_3_C_2_T*_x_* composites showed a hollow structure with a pore size of 5 μm. The influence of N-doped degrees on the electromagnetic interference (EMI) shielding performance was investigated over 8–18 GHz. Therefore, the controlled N-doping composites exhibited reflection-based EMI shielding performance due to the electrical conductivity and the special three-dimensional (3D) honeycomb-like structure. The achieved average EMI shielding values were 52.38 dB at the X-band and 72.72 dB at the K_u_-band. Overall, the Ag@N-Ti_3_C_2_T*_x_* foam, due to its special 3D honeycomb-like structure, not only meets the characteristics of light weight, but also exhibits ultra-high-efficiency EMI shielding performance, revealing great prospects in the application of electromagnetic wave shielding field.

## 1. Introduction

The development of telecommunication and portable electronic devices plays an important role in civilian and military applications; on the other hand, it has also been issued to cause serious electromagnetic interference problems. It may even threaten the human health. Hence, it is an urgent task to explore lightweight, efficient and environmentally friendly electromagnetic interference shielding materials to solve this kind of problem [1,2,3,4].

Traditionally, the efficient electromagnetic interference (EMI) shielding materials are made with intrinsic high electric conductivity or magnetics. Zeng et al. [5] synthesized Ni@carbon nanotubes (CNTs) in situ by the solvothermal method, and subsequently obtained PVDF/CNTs/Ni@CNTs composite films by solution casting and compression molding, which exhibited 51.4 dB of electromagnetic shielding performance. Younes et al. [6] coated carbon nanostructured (CNS) mats with Fe_3_O_4_ particles and explored its effect on electromagnetic shielding performance. Adding Fe_3_O_4_ to CNS mats, the electromagnetic shielding performance increased from 46.09 dB to 60.29 dB. However, due to the high density of metal and the shortcomings of easy agglomeration of carbon nanotubes, it does not meet the requirements for new electromagnetic shielding materials. Compared with traditional electromagnetic shielding materials, such as metals and carbon nanotubes, Ti_3_C_2_T*_x_* shows great prospects for EMI applications owing to its super high intrinsic electrical conductivity and chemically tunable properties. The EMI shielding performance of Ti_3_C_2_T*_x_* was first reported by Yury et al. [7] in 2016. They found that pure Ti_3_C_2_T*_x_* film exhibited EMI effectiveness of 92 dB in the range of 8.2–12.4 GHz. The 60 wt% Ti_3_C_2_T*_x_*/paraffin composites presented the EMI shielding performance of 39.1 dB [8]. However, the application of Ti_3_C_2_T*_x_* in the field of electromagnetic shielding is often limited by the self-stacking effect. By fabricating the lamellar structure into a 3D structure, the self-stacking effect can be improved. Meanwhile, the transmission path of electromagnetic waves in the porous network can be increased, thus enhancing the electromagnetic wave loss [9]. When Zhang et al. [10] transferred the MXene film into MXene foam through the hydrazine-induced foaming method, the EMI shielding performance was improved from 53 dB to 70 dB. The increasing EMI shielding values were due to the high attenuation efficient of the electromagnetic wave in the three-dimensional cellular MXene foam. Wu et al. [11] prepared lightweight MXene/sodium alginate (SA) aerogel and obtained an EMI shielding performance of 70.5 dB.

These findings have prompted the exploration of MXene 3D structured composites to further enhance the intrinsic EMI shielding performance of MXene through building 3D microporous structures [12]. Based on previous work, 3D MXene based composites cannot only improve the electromagnetic shielding performance but also exhibit a good mechanical properties and high stability [13]. For example, the EMI shielding performance of 3D MXene/reduced graphene oxide (rGO)/polyurethane containing Diels-Alder bounds (PUDA) composite hardly changed after 5000 bending cycles [14]. MXene/AgNWs/Epoxy aerogel material obtained an EMI shielding performance of 94.1 dB and exhibited excellent thermal conductivity [15].

Recently, researchers reported that increasing the electrical conductivity of MXene through nitrogen doping could advance the electromagnetic absorption performance [16]. For example, a specific capacitance of Ti_3_C_2_T*_x_* (201 F/g) could be increased by nitrogen doping to 340 F/g [16]. Li et al. [17] obtained −59.20 dB absorption performance of N-doped Ti_3_C_2_T*_x_* at 10.56 GHz by N_2_ plasma treatment method on Ti_3_C_2_T*_x_*. Since the absorption loss is essential to the electromagnetic shielding performance, we intended to explore the electromagnetic shielding performance of N-doped 3D MXene based composites.

Inspired by these studies and based on our previous work [18,19], we employed a facile and stable route to fabricate a honeycomb-like N-doped Ag@Ti_3_C_2_T*_x_* foam. While not destroying its microscopic morphology, high electrical conductivity can also be obtained and, at the same time, it can meet the characteristic of light weight. The uniform distribution of Ag particles and successful doping of nitrogen could significantly increase the conductivity of Ti_3_C_2_T*_x_* MXene. Meanwhile, the constructed 3D cellular structure contributes to multiple reflections of electromagnetic wave in the inside channels. In terms of these effects, 3D Ag@N-Ti_3_C_2_T*_x_* foam exhibited an improved electromagnetic shielding property (72.72 dB at the K_u_-band). This provides a new prospect for the development of advanced EMI shielding materials.

## 2. Experimental

### 2.1. Materials

The materials used in this work were listed in Table 1. All of them were used without further purification.

### 2.2. Synthesis of Honeycomb-like Ag@Ti_3_C_2_T_x_ Composites

Ti_3_C_2_T*_x_* MXene suspension was prepared using the etching method as previous reported [20]. In the process of preparing honeycomb-structured Ag@Ti_3_C_2_T*_x_* composites, the Ti_3_C_2_T*_x_*/PMMA pellets were first fabricated by adding 0.04 g of PMMA (5 μm) into a 40-mL Ti_3_C_2_T*_x_* suspension (1 mg/mL) and stirring for 1 h at room temperature. Subsequently, 0.1 M NaOH was added to the mixture until the pH value was 11. Then, 20 mL of AgNO_3_ solution (1.8 mM) was gradually added and stirred for 30 min. Next, the mixture was heated in a microwave (200 W) for 1 min. Subsequently, the mixture was cooled at room temperature, washed and filtered through a microporous monolayer membrane (Celgard 3501, 11 Technology Co. Ltd. Changchun). Finally, to remove the PMMA template, the composite membrane was annealed at 400 °C for 0.5 h in an argon atmosphere, and the self-supporting honeycomb-structured Ag@Ti_3_C_2_T*_x_* composites were obtained [19].

### 2.3. Synthesis of 3D Honeycomb-like Ag@N-Ti_3_C_2_T_x_ Foam

To investigate to effect of nitration degree on the physical property of MXene, the as-prepared honeycomb Ag@Ti_3_C_2_T*_x_* samples were immersed into 40 mL of hydrazine hydrate in an oil bath at 75 °C and stirred for 0 h, 12 h, 18 h and 24 h, respectively. During this process, hydrazine hydrate decomposed into NH_3_ that would replace the functional groups on the surface of Ag@Ti_3_C_2_T*_x_*. The obtained suspension was centrifuged at 5000 rpm for 5 min, and this process was repeated three times. The final products of Ag@N-Ti_3_C_2_T*_x_* were obtained by vacuum filtration and drying.

### 2.4. Characterization

The crystal structure of Ag@N-Ti_3_C_2_T*_x_* composites was characterized using X-ray diffraction (XRD) coupled with Cu-Kα radiation (λ = 0.15418 nm). The scanning electron microscopy (SEM, JSM-7200F, JEOL, Tokyo) and transmission electron microscope (TEM, FEI Talos F200X, Waltham, America) were used to observe the microscopic morphologies and structures of samples. The surface chemical composition and valence states were measured using a X-ray photoelectron spectroscopy (XPS) spectrometer (Thermo Scientific Escalab 250Xi, Waltham, America). The conductivity of the composites was measured by a four-probe tester (RTS-8).

The paraffin-based Ag@N-Ti_3_C_2_T*_x_* composites with a mass fraction of 50% were prepared for the evaluation of their EMI shielding performance. They were pressed into a ring shape with an inner distance of 3.04 mm and an outer diameter of 7.00 mm. The EMI shielding performance was calculated using the scattering parameters tested by the Vector network analyzer (MS46322B, Anli Co., Ltd., Kanagawa, Japan) at X-band (8–12 GHz) and K_u_-band (12–18 GHz). All the formulas used for calculation of the EMI shielding performance could be found in the Appendix A [21].

## 3. Results and Discussion

The fabrication process of the 3D honeycomb-like Ag@N-Ti_3_C_2_T*_x_* foam is shown in Figure 1. Based on our previous work [18,19], the prepared honeycomb-like Ag@Ti_3_C_2_T*_x_* composites were further nitrogen doped in an oil bath with hydrazine hydrate as a nitrogen source. During this process, hydrazine hydrate decomposed into NH_3_ that would replace the functional groups on the surface of Ag@Ti_3_C_2_T*_x_*, and the Ag@N-Ti_3_C_2_T*_x_* foam was successfully prepared.

The XRD patterns of the prepared composites are shown in Figure 2a. The results showed that the (002) characteristic peak of Ti_3_C_2_T*_x_* was slightly left shifted from 6.42° to 5.23°. In the N-doped treatment of Ti_3_C_2_T*_x_* was further left shifted [16]. For the Ag@N-Ti_3_C_2_T*_x_* composites, the peak at 38° corresponds to the (111) crystal plane of fcc Ag. The surface chemical information was recorded in the XPS spectra of Figure 2b. As shown in the survey spectra, we could detect the elemental signals of F, Ti, O, C, Ag and N from the Ag@N–Ti_3_C_2_T*_x_* composites. Compared to pure Ti_3_C_2_T*_x_*, the characteristic peak of N 1s at around 400 eV appears after the N-doped treatment. The reason for the appearance of N-Ti_3_C_2_T*_x_* is that hydrazine hydrate decomposes to produce NH_3_, and the functional groups on the surface of Ti_3_C_2_T*_x_* react with NH_3_ or are replaced. Thus, during the doping process, part of the NH_3_ is attached to the surface of Ti_3_C_2_T*_x_* to form the N-Ti_3_C_2_T*_x_*. The signal of N 1s from the N-doped honeycomb-like Ag@N-Ti_3_C_2_T*_x_* is weaker due to the presence of the Ag peak at 370 eV. The above results demonstrate the successful N-doping and preparation of honeycomb-like structured Ag@N-Ti_3_C_2_T*_x_* composites.

The high-resolution XPS spectra of Ti 2p, C 1s, O 1s, N 1s and Ag 3d are given in Figure 3. With the N-doping, it shows Ti-N bonds at the binding energies of 456 eV and 462 eV (Figure 3a) [16]. In Figure 3b, the peaks centered at 282 eV, 283 eV, 285 eV, 287 eV and 289 eV, corresponding to C-Ti-T*_x_*, C-C, C-N, C-O and C=O, respectively [22]. The deconvoluted peaks of the O 1s at 530 eV, 531 eV, 532 eV and 535 eV are assigned to Ti-O-Ti, Ti-OH, O-N and -COO, respectively (Figure 3c). It can be seen that after N-doping, there were the C-N and O-N peaks [23], and the five deconvoluted peaks at 396.8 eV, 398.5 eV, 399.6 eV, 401.2 eV and 402.4 eV in the N 1s spectrum are assigned to N-Ti, pyridine-N, C-NH, C-NH_2_ and oxidized-N, respectively (Figure 3d). The above analyses further indicate the successful doping of N in Ti_3_C_2_T*_x_* foam. Moreover, the high resolution of Ag 3d spectrum is displayed in Figure 3e. Compared to the N-Ti_3_C_2_T*_x_* composites (Appendix A), the two peaks at the binding energies of 368 eV and 374 eV, linked to the formation of Ag° 3d_3/2_ and Ag° 3d_5/2_, respectively, which further confirmed that the Ag particles exist in the composites.

The morphologies of N-Ti_3_C_2_T*_x_* lamellae and honeycomb-like Ag@N-Ti_3_C_2_T*_x_* foam were characterized by SEM, as shown in Figure 4a–e. There are wrinkles in the N-Ti_3_C_2_T*_x_*, which is due to the lattice distortion caused by N-doping in the pristine Ti_3_C_2_T*_x_* layer [24]. The prepared honeycomb-like Ag@Ti_3_C_2_T*_x_* was nitrided in hydrazine hydrate to obtain honeycomb-like Ag@N-Ti_3_C_2_T*_x_* foam, and the honeycomb-like structure was retained even after N-doping. The measured pore diameter of Ag@N-Ti_3_C_2_T*_x_* composites was about 4 μm, and the diameter of an N-Ti_3_C_2_T*_x_* thin shell was about 10 nm (Figure 4b–e).

To further observe the elemental distribution in the samples, an energy spectrum surface scan of N-Ti_3_C_2_T*_x_* was performed. As shown in Figure 4f–i, the EDS mapping confirmed the uniform distribution of Ti, C, O and N elements in the N-Ti_3_C_2_T*_x_* composite, which indicates the successful doping of N elements as well.

The microstructure of N-Ti_3_C_2_T*_x_* and Ag@N-Ti_3_C_2_T*_x_* foam was further characterized by TEM. As shown in Figure 5a, the N-Ti_3_C_2_T*_x_* retains the flake-like structure of two-dimensional materials. The high-resolution TEM (HRTEM) image (Figure 5b) of the N-Ti_3_C_2_T*_x_* composites showed the corresponding lattice fringes from (006) of Ti_3_C_2_T*_x_* [25]. The SAED diagram (Figure 5c) confirms the hexagonal structure of parent Ti_3_C_2_T*_x_* MXene phase was maintained in the N-Ti_3_C_2_T*_x_*. In Figure 5d, the original honeycomb-like structure had not collapsed even after being N-doped, and the Ag nanoparticles were closely attached to the Ti_3_C_2_T*_x_* shell (Figure 5d). The HRTEM image (Figure 5e) presents the corresponding lattice fringes of (111) of Ag and (006) of N-Ti_3_C_2_T*_x_*. Figure 5f presents a variety of diffraction rings, confirming that the polycrystalline phase at the interface of the Ag and Ti_3_C_2_T*_x_* and the spacing of diffraction rings correspond to the (111) of Ag and (100) (110) of Ti_3_C_2_T*_x_*, respectively.

The elemental distribution on the surface of the prepared honeycomb-like structured Ag@N-Ti_3_C_2_T*_x_* 5 μm was shown in Figure 5h–l. The EDS pattern confirms the coexistence of Ti, C, O, N and Ag components in Ag@N-Ti_3_C_2_T*_x_* 5 μm composites, and the distribution of each element is relatively uniform, which further confirms the N doping in Ag@N-Ti_3_C_2_T*_x_* composites.

As described in the introduction, the conductivity of materials is crucial to the EMI shielding performance. We measured the electrical conductivity of the Ti_3_C_2_T*_x_*, N-Ti_3_C_2_T*_x_* and Ag@N-Ti_3_C_2_T*_x_* with N-doping treated at varying hours (Figure 6). The N-doped Ti_3_C_2_T*_x_* has an electrical conductivity of 350 S/cm, which is about 1.7 times higher than that of pure Ti_3_C_2_T*_x_*. In the presence of silver particles, the electrical conductivity of the honeycomb-like Ag@Ti_3_C_2_T*_x_* composites was 500 S/cm. As nitriding Ag@Ti_3_C_2_T*_x_* composites for 12 h, 18 h and 24 h, the electrical conductivity of the honeycomb-like Ag@N-Ti_3_C_2_T*_x_* composites was 520, 570 and 540 S/cm, respectively. This is mainly due to the great conductivity of sliver and the electron-giving effect of element N [26]. However, the silver effect was much more significant than the nitrogen doping effect. Moreover, as extra N doped into Ti_3_C_2_T*_x_*, the increasing internal free electron concentrations indeed increase the conductivity of Ti_3_C_2_T*_x_* nanosheets [27], the over doping of nitrogen may make the Ti_3_C_2_T*_x_* oxidized and then generate TiO_2_.

The frequency dependent EMI shielding performance of the Ag@N-Ti_3_C_2_T*_x_* composites was shown in Figure 7a. The average EMI SE total of pure Ti_3_C_2_T*_x_*, N-Ti_3_C_2_T*_x_* and honeycomb-like structured Ag@Ti_3_C_2_T*_x_* 5 μm at the X-band was 23.50 dB, 30.54 dB and 51.15 dB, respectively, while at the K_u_-band, the values were 30.33 dB, 31.97 dB and 56.64 dB, respectively. With the N-doping treatment at 18 h, the average EMI SE total of the Ag@N-Ti_3_C_2_T*_x_* 5 μm honeycomb-like was 52.38 dB over the X-band and 72.72 dB over the K_u_-band, which was higher than other samples in the frequency range of 10–18 GHz (Appendix A). In addition, its reflectivity was the lowest in the K_u_-band compared to other absorbers (Appendix A). This phenomenon may be due to the N-doping and the electric conductivity.

In addition, at 12 GHz the SE totals of Ti_3_C_2_T*_x_*, N-Ti_3_C_2_T*_x_* and 18 h Ag@N-Ti_3_C_2_T*_x_* 5 μm were 26.41 dB, 31.27 dB and 60.10 dB, respectively (Figure 7c), while at 18 GHz the SE total values of Ti_3_C_2_T*_x_*, N-Ti_3_C_2_T*_x_* and 18 h Ag@N-Ti_3_C_2_T*_x_* were 29.96 dB, 33.91 dB and 75.27 dB, respectively (Figure 7d). Overall, the 18 h Ag@N-Ti_3_C_2_T*_x_* 5 μm composites exhibit the best EMI shielding performance with an average EMI SE total of 64.58 dB. The improvement in absorption loss has significantly contribution to SE_T_ with a negligible improvement in reflection loss.

The electromagnetic shielding mechanisms of the honeycomb-like Ag@N-Ti_3_C_2_T*_x_* composite are summarized in Figure 7b. As fabricating the two-dimensional Ti_3_C_2_T*_x_* into a three-dimensional honeycomb-like structure, the propagation paths of electromagnetic waves diameter increase [28]. The transmissivity of pure Ti_3_C_2_T*_x_* was obviously decreased (Appendix A). The multiple reflections of electromagnetic waves occurring in the honeycomb-like foam thus improve the electromagnetic shielding performance of Ag@N-Ti_3_C_2_T*_x_* composites [29]. Moreover, the special honeycomb-like structure provides high specific surface area. The measured values for N-Ti_3_C_2_T*_x_* and Ag@N-Ti_3_C_2_T*_x_* are 49.89 and 67.47 m^2^/g, respectively, and the Ag nanoparticles distributed at the interfaces could generate the interfacial polarization relaxation loss. The charges then accumulate at these interfaces, leading to a strong interfacial polarization effect. The other factor can be attributed to the conduction loss. The conduction loss is related to the conductivity of the material. The higher the conductivity, the greater the macroscopic current caused by the carriers (including the current caused by the change in the electric field and the eddy current caused by the change of the magnetic field), which is conducive to the conversion of electromagnetic energy into heat energy. Both the introduction of Ag particles and the N-doping increase the electrical conductivity of the MXene material [26,30,31]. The increasing conductivity makes the skin depth ($\delta ={\left(\sqrt{\pi f\mu \sigma }\right)}^{-1}$) become smaller, thus, the electromagnetic wave cannot penetrate the material, and then electromagnetic shielding performance can also be improved. Hydrazine hydrate provides a reducing environment as a nitrogen source to avoid oxidation. However, the long nitriding time would lead to the generation of NH_3_ gas from hydrazine hydrate and cause the Ti_3_C_2_T*_x_* to be oxidized into TiO_2_. The generation of TiO_2_ will lead to a decrease in its own conductivity and affect the electromagnetic shielding performance of the material.

## 4. Conclusions

In this research, honeycomb-like Ag@N-Ti_3_C_2_T*_x_* with different degrees of nitridation was prepared by using hydrazine hydride nitridation. Even after N-doping, the Ag@N-Ti_3_C_2_T*_x_* composite still retained its original honeycomb-like structure in the same size. In the nitridation for 18 h, the electrical conductivity of Ag@N-Ti_3_C_2_T*_x_* composites was 500 S/cm. The electromagnetic shielding performance was significantly improved by N-doping and the decoration of Ag particles. The average electromagnetic shielding performance was 52.38 dB, covering the whole X-band, and 72.72 dB over the Ku-band. The excellent electromagnetic shielding property of Ag@N-Ti_3_C_2_T*_x_* composites comes from its own conductivity loss, the interfacial polarization between Ag nanoparticles and the N-Ti_3_C_2_T*_x_* shell, the dipole polarization at the N dipole polarization at the defects, and multiple reflections and scatterings caused by the hollow honeycomb-like structure of the composite.

## Figures and Tables

**Figure 1 nanomaterials-12-02967-f001:**
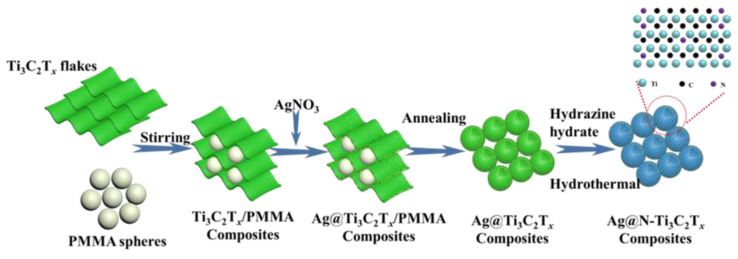
Schematic diagram of the preparation process of Ag@N-Ti_3_C_2_T*_x_* composites with honeycomb-like structure.

**Figure 2 nanomaterials-12-02967-f002:**
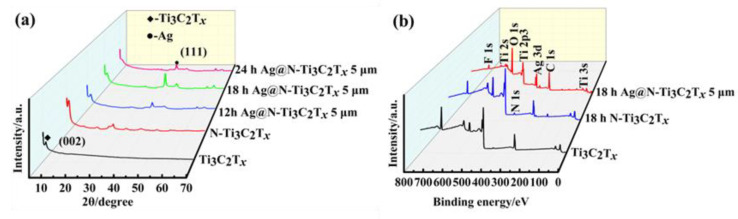
(**a**) XRD patterns and (**b**) XPS survey of Ti_3_C_2_T*_x_*, N-Ti_3_C_2_T*_x_* and Ag@N-Ti_3_C_2_T*_x_* composites.

**Figure 3 nanomaterials-12-02967-f003:**
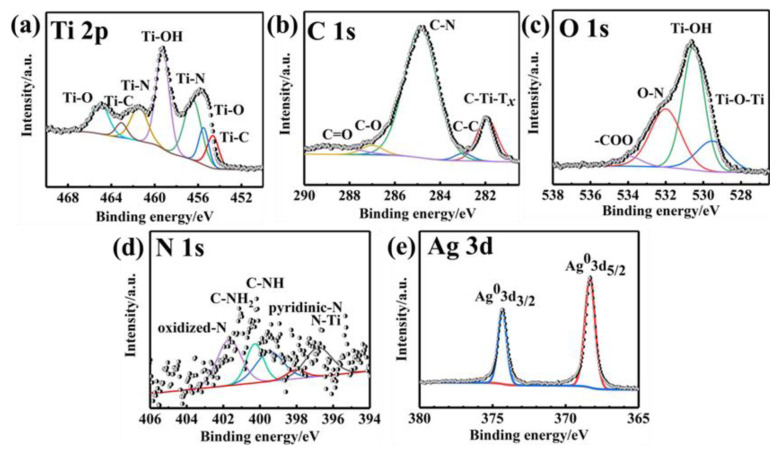
High-resolution XPS spectra of Ti 2p (**a**), C 1s (**b**), O 1s (**c**), N 1s (**d**) and Ag 3d (**e**) of Ag@N-Ti_3_C_2_T*_x_* composites.

**Figure 4 nanomaterials-12-02967-f004:**
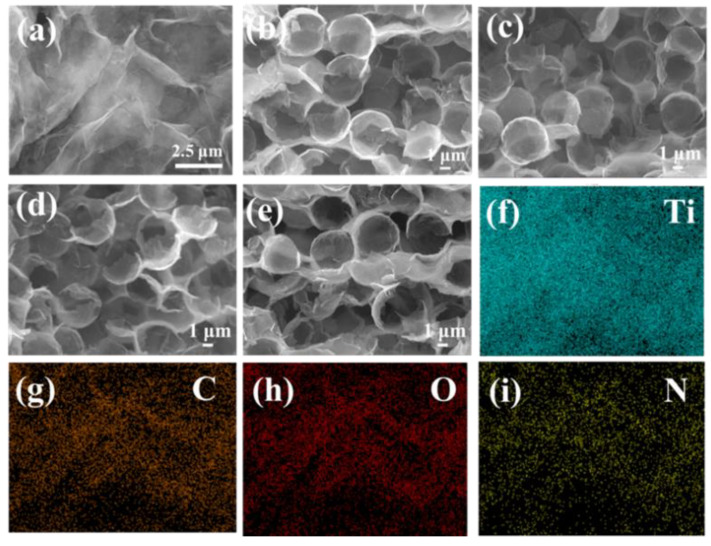
SEM images of (**a**) nitrogen doping Ti_3_C_2_T*_x_*, (**b**) Ag@Ti_3_C_2_T*_x_* foam, (**c**) 12h Ag@N-Ti_3_C_2_T*_x_* foam nitriding for 12 h, (**d**) Ag@N-Ti_3_C_2_T*_x_* foam nitriding for 12 h and (**e**) Ag@N-Ti_3_C_2_T*_x_* foam nitriding for 12 h. (**f**–**i**) EDS mappings of Ti, C, O and N elements, respectively.

**Figure 5 nanomaterials-12-02967-f005:**
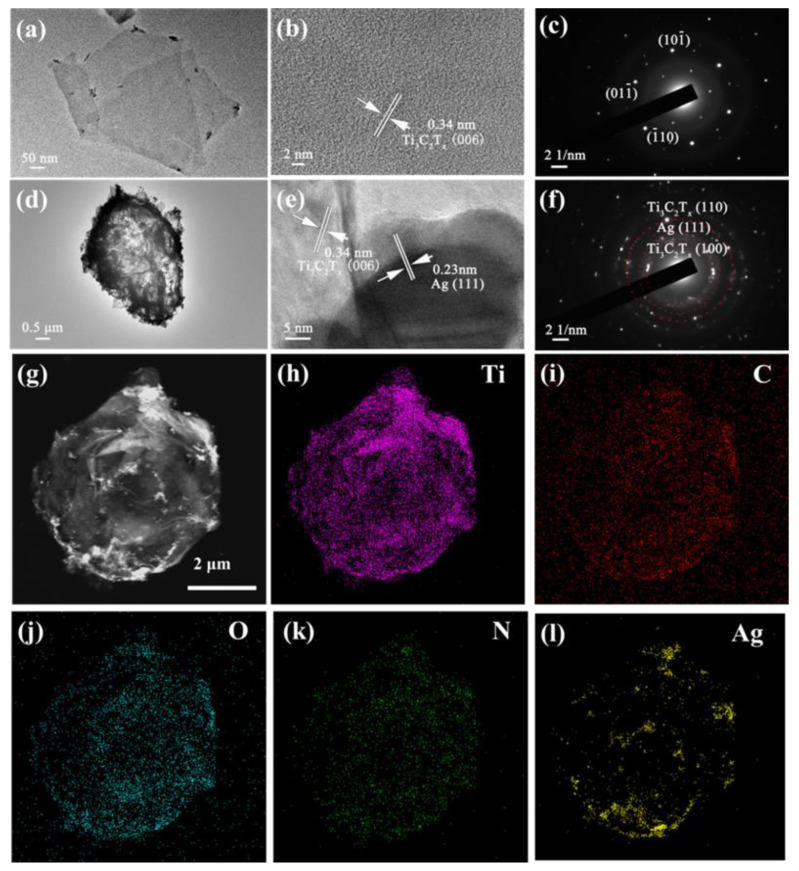
TEM image of (**a**) N-Ti_3_C_2_T*_x_* flakes, (**b**) HRTEM image of N-Ti_3_C_2_T*_x_* flakes, (**c**) SAED patterns of N-Ti_3_C_2_T*_x_* flakes, (**d**) TEM image of Ag@N-Ti_3_C_2_T*_x_* 5 μm, (**e**) HRTEM image of Ag and Ti_3_C_2_T*_x_* and (**f**) SAED patterns of Ag and Ti_3_C_2_T*_x_*. (**g**) TEM image of Ag@N-Ti_3_C_2_T*_x_*, (**h**–**l**) EDS mappings of Ti, C, O, N and Ag elements.

**Figure 6 nanomaterials-12-02967-f006:**
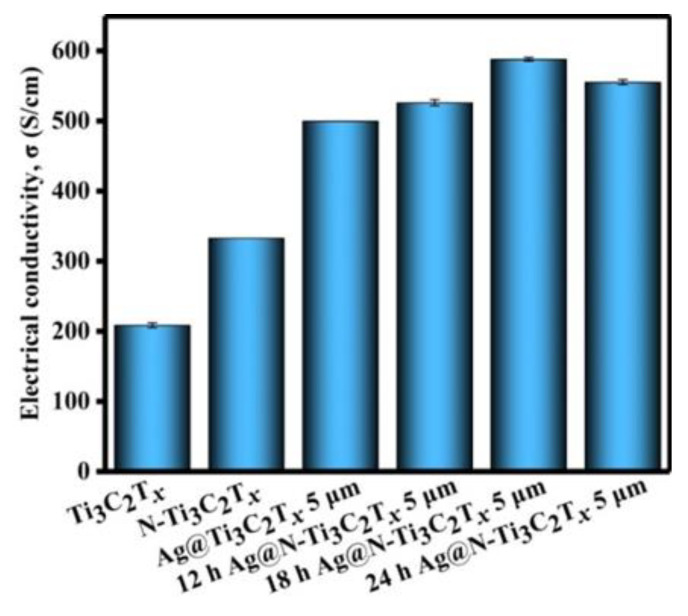
Electrical conductivity of Ti_3_C_2_T*_x_*, N-Ti_3_C_2_T*_x_* and Ag@N-Ti_3_C_2_T*_x_* composites. The error bars were from five measurements.

**Figure 7 nanomaterials-12-02967-f007:**
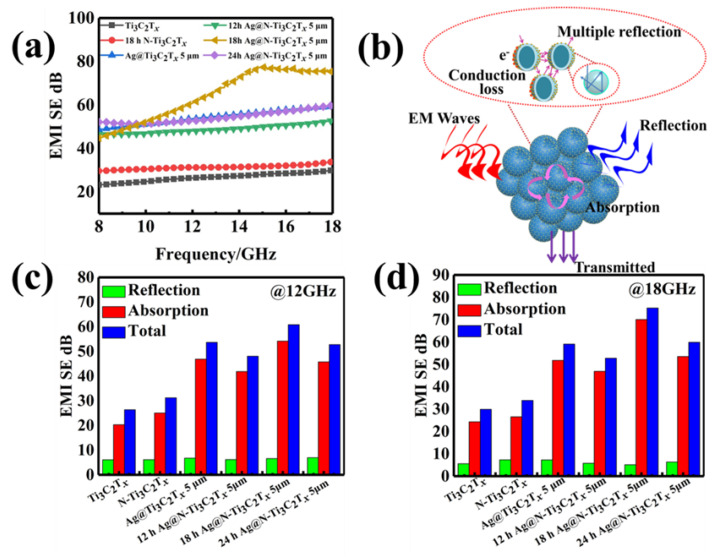
(**a**) EMI SE curve; (**b**) EMI shielding mechanism diagram of Ag@N-Ti_3_C_2_T*_x_* honeycomb structure; SE_R_, SE_A_ and SE_T_ of the composites at 12 GHz (**c**) and at 18 GHz (**d**).

**Table 1 nanomaterials-12-02967-t001:** List of materials used in this work.

Materials	Purity/Grain Size	Manufacturers
Ti_3_AlC_2_ MAX	≥99%	11 Technology Co., Ltd.
Hydrochloric acid (HCl)	36–38 wt%	Aladdin Biochemical Technology Co., Ltd.
Lithium fluoride (LiF)	≥99%	Macklin Biochemical Co., Ltd.
Polymethyl methacrylate (PMMA)	5 μm	Dongguan Kemai New Material Co., Ltd.
Ethanol	analytical reagent	Aladdin Biochemical Technology Co., Ltd.
Silver nitrate (AgNO_3_)	≥99.8%	Sinopharm Chemical Reagent Co. Ltd.
Sodium hydroxide (NaOH)	≥96%	Macklin Biochemical Co., Ltd.
Ammonia (N_2_H_4_·H_2_O)	≥98%	Aladdin Biochemical Technology Co., Ltd.

## Data Availability

Not Applicable.

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
