# Peer review of "N-Doped Honeycomb-like Ag@N-Ti3C2Tx Foam for Electromagnetic Interference Shielding"

_nanomaterials, 2022, doi:10.3390/nano12172967_

Round 1

Reviewer 1 Report

Please introduce, explain, clarify the notation used to indicate constituents within the foam under study. 

It is attractive to call this foam a "honeycomb" structure. this may or may not be appropriate due to the lack of order and regularity in the structural images shown in this work. Please refer to the nomenclature used in foam related research articles. 

Please clarify N-doping as Nitrogen doping. N-type doping is completely different in semiconductor applications and can be confusing without clarification. 

Page 1 line 43: please state the wavelength ranges associated with the shielding performance of the listed products.

Page 2 line 53: Use more quantitative language in place of words like "hardly"

Page 2 lines 62 and 65: Please introduce and spell out all acronyms at the point of first use. 

Table 1: What does AR mean in regards to the purity of ethanol. please clarify. Was it reagent grade or denatured?

Line 127: What is the origin of Fluorine in your analysis data?

Figure 4 and Figure 5: the images were listed out of order in the figure caption. They were also discussed within the text out of order of presentation. Please rearrange the order of the images or the order in which they are discussed in the text. 

Figure 6: What do the error bars represent and how many data points does each location average together. 

Figure 7b: Does scattering play any role in the shielding process? please clarify as to whether it does or does not. 

Reviewer 2 Report

It seems to me that the manuscript could be published. It seems to me that the manuscript has the ingredients to be worth of publication: a problem, its state of the art, an original approach to solve, or improve the solution to the problem, some experimental results, some theory. The manuscript seems to me well written. I am not aware of ways to further improve the manuscript, I did not find things that would need improvement.

Reviewer 3 Report

N-doped honeycomb Ag@N-Ti3C2Tx foam for electromagnetic  interference shielding

This manuscript discusses the fabrication of Ag@N-Ti3C2Tx foam doped with Nitrogen and uses this fabricated structure for the EMI shielding application.

Comments:

The abstract and introduction need to be rewritten.

1.       I believe that Ti3C2Tx is MXene. Why don’t you mention that? That is strange

2.       Abstract needs to be thoroughly revised as it is weak and has some English problems.

3.       Introduction also needs to be thoroughly revised as there are many grammars mistakes

e.g.  

1)      “The development of telecommunication and portable electronic devices plays” should be: play,

2)      “ in the application of civilian and military” should be in civilian and military applications

3)      Line 32-33, Traditionally and generally; please use one of the following

4)      Line 33 Among them, them refers to what; you didn’t mention any

5)      Line 43, is it MXene foams or MXene foam?

4.       According to your literature review line 43 ref [8] Zhang et al. has already done the work that you did, so what are you trying to do different than him? However, he obtained better results than your results

5.       Why did use cannot line 51?  I am confused

6.       Rewrite line 56-57

7.       Line 57, “According to recent studies, N doping may be” delete may be

8.       The motivation and the novelty of the work need to be highlighted

9.       More literature should be included to cover reports about thin films, CNTs mats, honeycomb structures etc. such as

https://doi.org/10.1021/acsami.0c00034

https://doi.org/10.1016/j.synthmet.2021.116731

https://doi.org/10.1016/j.compositesb.2021.109500

https://doi.org/10.1016/j.synthmet.2019.04.026

https://doi.org/10.1016/j.carbon.2019.08.024

10.   The synthesis process of the  honeycomb Ag@Ti3C2Tx should be clearly described in the manuscript and detailed scheme should be provided

11.   line 90 you mentioned the following reaction. However, I don’t see any reaction

12.   line 92-93 were obtained via,

13.   line 192, you just mentioned in the introduction that the conductivity of materials is crucial to the EMI shielding performance. however, you didn’t explain why. Could you please explain why?

14.   Would you please explain why you used Ag

15.   Why did you stir for different time

16.   Could you revise the caption of figure 4

Results and discussion

17.   Again I think English for the whole manuscript needs to be checked

18.   Figure 7 C and D are hard to follow as X axes labels are so crossed

19.   18 h Ag@N-Ti3C2Tx 5 μm sample has higher electrical conductivity?

20.   Why the reflection of the 12 h Ag@N-Ti3C2Tx 5 μm is higher than other?

21.   What is the effect of time

22.   According to your results the EMI shielding performance of 18 h Ag@N-Ti3C2Tx 5 μm is higher than others and match with electrical conductivity, could you please explain this better and show the physics that support your findings

23.   Line 212-220, you don’t need to mention all numbers it makes hard to follow. You could describe in general and refer to the figure.

24.   Line 226 “Moreover, the special honeycomb structure provides high specific surface area” did you do surface area analysis can you provide BET

25.   Line 227 And

26.   In3 229-230 “The other main mechanism can be attributed to the conduction loss.” Could you please provide some explanation and evidence?

more details and discussion need to be added  

Round 2

Reviewer 3 Report

I think it should be accepted now